# MetaEmbed: Scaling Multimodal Retrieval at Test-Time with Flexible Late Interaction

**Zilin Xiao[1,2], Qi Ma[1], Mengting Gu[1], Jason Chen[1], Xintao Chen[1], Vicente Ordonez[2], Vijai Mohan[1]**
[1]Meta Superintelligence Labs    [2]Rice University
zilin@meta.com

## Abstract

Universal multimodal embedding models have achieved great success in capturing semantic relevance between queries and candidates. However, current methods either condense queries and candidates into a single vector, potentially limiting the expressiveness for fine-grained information, or produce too many vectors that are prohibitive for multi-vector retrieval. In this work, we introduce MetaEmbed, a new framework for multimodal retrieval that rethinks how multimodal embeddings are constructed and interacted with at scale. During training, a fixed number of learnable Meta Tokens are appended to the input sequence. At test-time, their last-layer contextualized representations serve as compact yet expressive multi-vector embeddings. Through the proposed Matryoshka Multi-Vector Retrieval training, MetaEmbed learns to organize information by granularity across multiple vectors. As a result, we enable test-time scaling in multimodal retrieval where users can balance retrieval quality against efficiency demands by selecting the number of tokens used for indexing and retrieval interactions. Extensive evaluations on the Massive Multimodal Embedding Benchmark (MMEB) and the Visual Document Retrieval Benchmark (ViDoRe) confirm that MetaEmbed achieves state-of-the-art retrieval performance while scaling robustly to models with 32B parameters. Code is available at https://github.com/facebookresearch/MetaEmbed.

## 1 Introduction

Multimodal embedding models play an essential role in image search (Gordo et al., 2016), visual question answering (Hu et al., 2018; Zheng et al., 2021) and visual document retrieval (Faysse et al., 2025), where models project heterogeneous inputs into a unified representation space. While existing methods, including CLIP (Radford et al., 2021), BLIP (Li et al., 2022) and SigLIP (Zhai et al., 2023) have demonstrated superior performance in cross-modal retrieval, their performance remains limited in scenarios where the inputs involve complex and diverse instructions. Thanks to recent advances in building embeddings through foundation vision-language models (VLMs), one could apply contrastive learning on the extracted embedding from the hidden states of the last layer of a VLM to learn meaningful multimodal representations while retaining pre-trained knowledge.

Despite growing progress in multimodal embedding VLMs, the common practice of condensing the entire query and candidate into a single vector is not an optimal choice, as fine-grained details are lost between modalities (Yao et al., 2022; Thrush et al., 2022) and this process has theoretical limitations (Weller et al., 2025). In text retrieval, ColBERT (Khattab & Zaharia, 2020) introduced a multi-vector late interaction mechanism that retains multiple token-level embeddings and uses a lightweight scoring between query and document token representations. This approach preserves significantly more contextual information than single-vector methods while still allowing independent encoding of queries and documents, and has motivated a recent trend of devising multi-vector embeddings for multimodal retrieval (Faysse et al., 2025; Xu et al., 2025; Günther et al., 2025).

However, multi-vector methods incur substantial efficiency costs in terms of *index size*, *retrieval latency* and *feasibility*. In these methods, each image is encoded into hundreds of patch embeddings, and each query text into several token embeddings, all of which must be stored and compared during retrieval. This results in large index sizes and slower retrieval processing. Moreover, multimodal-to-

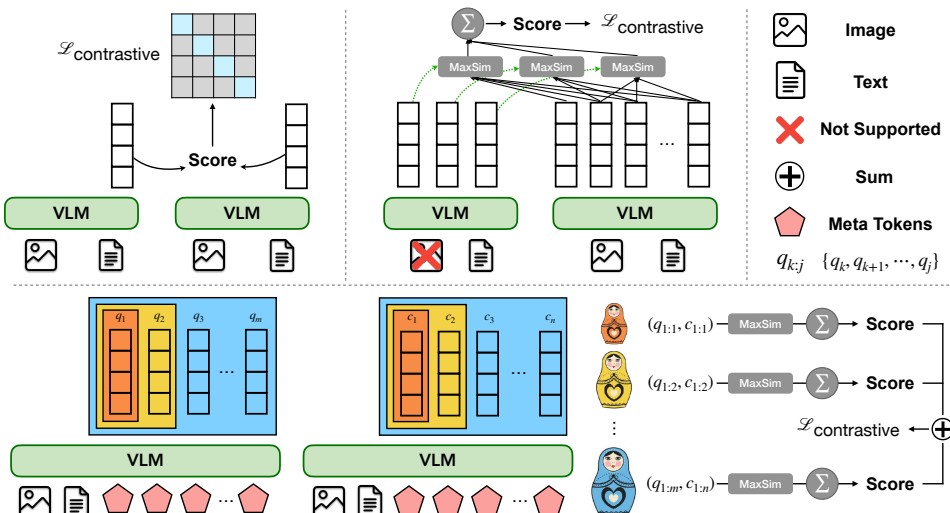

Figure 1: **Upper Left:** Single vector retrieval method computes a score for each pair of query and candidate and uses a contrastive objective to maximize the score for corresponding pairs. **Upper Right:** Multi-vector retrieval aggregates maximum similarities across vector pairs before training. **Lower:** METAEMBED structures query and candidate vectors into hierarchical nested groups and trains coarse-to-fine multi-vector embeddings that enable scalable and flexible retrieval.

multimodal retrieval becomes impractical when both the query and candidate sides contain images, as similarity computation for each query-candidate pair involves interactions between thousands of query tokens and thousands of candidate tokens, making both training and inference prohibitive due to computational demands.

In this work, we propose METAEMBED as a scalable late-interaction training recipe that advances multimodal retrieval with a flexible multi-vector method, illustrated in Figure 1. Instead of encoding the query and candidate into one vector, we introduce a small number of learnable *Meta Tokens* appended to the input sequence of the query and candidate. Their last-layer hidden states serve as a set of contextualized representations for late interaction, namely *Meta Embeddings*. To enable flexible late interaction, where users can trade off retrieval accuracy against computational budget and retrieval latency, we draw inspiration from Matryoshka Representation Learning (Kusupati et al., 2022) and design the Matryoshka Multi-Vector Retrieval (MMR) module in METAEMBED. By performing contrastive learning across parallel nested groups of representations **at training-time**, the model learns coarse-to-fine multi-vector embeddings that can be selectively utilized for late interactions depending on the computation budget **at test-time**. Increasing the number of *Meta Embeddings* used at indexing improves the retrieval quality at the cost of index storage budget and retrieval latency, thus enabling test-time scaling in multimodal retrieval.

We first validate METAEMBED on the Massive Multimodal Embedding Benchmark (MMEB) (Jiang et al., 2024) and Visual Document Retrieval Benchmarks (ViDoRe) v2 (Faysse et al., 2025; Macé et al., 2025), which represent a comprehensive suite of retrieval tests covering images, text and visual documents. Our experiments show that METAEMBED achieves state-of-the-art retrieval performance across diverse scenarios. To further examine its generality and training scalability, we evaluate METAEMBED with different VLM architectures and model sizes. Notably, test-time scalability remains effective even at 32B scale, with minimal diminishing returns as models grow larger. We hope METAEMBED charts a path toward multimodal retrieval systems that are both accurate and deployable at scale, advancing the pursuit of generality, efficiency, and flexibility.

## 2 RELATED WORK

**Multimodal Embedding.** These methods aim to project heterogeneous inputs into a shared representation space for cross-modal understanding and retrieval (e.g Frome et al. (2013); Kiros et al.

(2014); Faghri et al. (2018)). More recent large scale models such as CLIP (Radford et al., 2021), MetaCLIP (Xu et al., 2024; Chuang et al., 2025), BLIP (Li et al., 2022) and SigLIP (Zhai et al., 2023) encode each modality independently and apply contrastive training to enforce cross-modal alignment. More recent methods are built upon stronger VLMs (Xiao et al., 2024; Kong et al., 2025; Qin et al., 2025; Ju & Lee, 2025; Lin et al., 2025; Xiao et al., 2025). For instance, VLM2Vec (Jiang et al., 2025) adapts Phi-3.5-V (Abdin et al., 2024), VLM2Vec-V2 (Meng et al., 2025) and GME (Zhang et al., 2024b) builds on Qwen2 (Wang et al., 2024) and LLaVE (Lan et al., 2025) finetunes on LLaVA (Li et al., 2024). Beyond architectural choices, the community has also explored innovative strategies in data construction and training. MegaPairs (Zhou et al., 2024) and mmE5 (Chen et al., 2025b) curate large-scale synthetic data with sophisticated pipelines to support contrastive training. UniME (Gu et al., 2025) achieves strong results through diverse data combined with teacher model distillation. B3 (Thirukovalluru et al., 2025) incorporates novel insights into batch mining techniques. MoCa (Chen et al., 2025a) used continual pre-training to produce bidirectional embeddings. Nevertheless, many existing multimodal retrieval methods predominantly rely on single-vector retrieval, which hinders further scaling as embedding size becomes a bottleneck.

**Multi-Vector Retrieval.** Multi-vector retrieval refers to a family of dense retrieval methods that represent queries and documents with multiple embeddings rather than a single vector (Tolias et al., 2016; Tan et al., 2019; Ren et al., 2017), with ColBERT (Khattab & Zaharia, 2020) being a successful recent example of this paradigm by introducing a late interaction framework. While many variants have been proposed to improve multi-vector retrieval efficiency through approximation (Lee et al., 2023; Engels et al., 2023; Jayaram et al., 2024) and compression (Santhanam et al., 2022b;a; Li et al., 2023), naive ColBERT-style methods such as ColPali, ColQwen (Faysse et al., 2025) and others (Günther et al., 2025; Xu et al., 2025) still remain dominant in the context of text-image retrieval. However, these approaches do not support multimodal-to-multimodal retrieval, since introducing hundreds of image tokens on the query side renders both training and inference computationally prohibitive, highlighting the need for our proposed METAEMBED framework.

**Matryoshka Representation Learning.** Matryoshka Representation Learning (MRL) (Kusupati et al., 2022) was introduced to encode features at multiple granularities within a single vector in a nested structure. Popular text-only single-vector retrieval models (Zhang et al., 2025; Günther et al., 2025) natively support MRL, enabling retrieval to dynamically select the first few dimensions according to the available computational budget. Although prior work (Cai et al., 2025) has applied Matryoshka methods for token budgeting in VLM generation, to the best of our knowledge, METAEMBED presents the first work that leverages such a framework for multi-vector retrieval and achieves successful test-time scaling.

## 3 METHODOLOGY

In this section, we first revisit the definition of multimodal retrieval and how late interaction works to utilize multiple vectors for retrieval. Then we introduce the METAEMBED recipe, its model architecture, and how it enables test-time scaling in multimodal retrieval.

### 3.1 PRELIMINARIES

**Problem Definition.** Multimodal retrieval consists of retrieving relevant content across different modalities, where the query $\mathbf{q}$ can be text $q_t$, an image $q_i$ or a combination of both $(q_t, q_i)$. And the retrieval candidates $\mathbf{c}$ can likewise be of any modality or multimodal combination. Given a query $\mathbf{q}$ and a set of $\mathbf{N}$ candidates $\{\mathbf{c}_1, \mathbf{c}_2, \ldots, \mathbf{c}_N\}$, a multimodal retrieval model typically defines a similarity function $s(\mathbf{q}, \mathbf{c})$ to measure the relevance between $\mathbf{q}$ and a candidate $\mathbf{c}$. The retrieved top-1 prediction $\mathbf{c}^\star$ is then determined by:

$$\mathbf{c}^\star = \underset{\mathbf{c} \in \{\mathbf{c}_1, \ldots, \mathbf{c}_N\}}{\arg\max} \; s(\mathbf{q}, \mathbf{c}). \tag{1}$$

**Late Interaction.** For a query $\mathbf{q}$ and a candidate $\mathbf{c}$, let their multi-vector representations be denoted as $\mathbf{E_q} \in \mathbb{R}^{N_q \times D}$ and $\mathbf{E_d} \in \mathbb{R}^{N_d \times D}$, where $D$ is the embedding dimension, and $N_q, N_d$ are the number of token-level vectors for the query and the candidate, respectively. The late interaction

operator $\mathbf{LI}(q, d)$ captures the most informative alignment by selecting, for each query vector $\mathbf{E}_{\mathbf{q}}^{(i)}$, its maximum similarity (dot product) with the document vectors $\mathbf{E}_{\mathbf{d}}^{(j)}$, and summing across all query vectors:

$$\mathbf{LI}(q, d) = \sum_{i=1}^{N_q} \max_{j \in [1, N_d]} \left\langle \mathbf{E}_{\mathbf{q}}^{(i)}, \mathbf{E}_{\mathbf{d}}^{(j)} \right\rangle. \tag{2}$$

### 3.2 OUR DESIGN

**METAEMBED Recipe.** METAEMBED is designed as a scalable late-interaction retrieval model that introduces a small number of learnable *Meta Tokens* appended to the input sequence of both queries and candidates. These *Meta Tokens* are processed jointly with the original input by an underlying Vision-Language Model (VLM), and their final hidden states serve as *Meta Embeddings*. Unlike patch- or token-level embeddings, *Meta Embeddings* provide a set of compressed yet expressive vectors that capture fine-grained semantics through contextualization. This design drastically reduces the number of vectors required for retrieval while maintaining retrieval quality.

Formally, let a VLM with parameters $\boldsymbol{\theta}$ define a conditional probability distribution $p_{\boldsymbol{\theta}}(\mathbf{y} \mid \mathbf{x}, \mathcal{I})$ where $\mathbf{x} = [x_1, \ldots, x_n]$ is the query or document text prompt and $\mathcal{I}$ are associated input images. METAEMBED augments the input with learnable *Meta Tokens*: queries use $\mathbf{M}_q \in \mathbb{R}^{R_q \times D}$, and candidates use $\mathbf{M}_c \in \mathbb{R}^{R_c \times D}$. For an input $(\mathbf{x}, \mathcal{I})$, the transformer consumes $\mathbf{z}^{(0)} = \left[ \mathbf{v} ; \mathbf{t} ; \mathbf{M}_q ; \mathbf{M}_c \right] \in \mathbb{R}^{(P+n+R_q+R_c) \times D}$, where $\mathbf{v}$ and $\mathbf{t}$ are $P$ visual patches tokens and text inputs. The model produces last-layer hidden states $\mathbf{H} = F_{\boldsymbol{\theta}}(\mathbf{z}^{(0)}) \in \mathbb{R}^{(P+n+R_q+R_c) \times D}$, where $F_{\boldsymbol{\theta}}$ denotes the transformer network parameterized by $\boldsymbol{\theta}$. We extract the final hidden states at the *Meta Tokens* positions to obtain query-side embeddings $\mathbf{E}_{\text{meta}}^{(q)} \in \mathbb{R}^{R_q \times D}$ or candidate-side embeddings $\mathbf{E}_{\text{meta}}^{(c)} \in \mathbb{R}^{R_c \times D}$, followed by L2 normalization. Each $\mathbf{E}_{\text{meta}}^{(q)}$ and $\mathbf{E}_{\text{meta}}^{(c)}$ constitutes a compact, contextualized *multi-vector* representation produced in two separate forward passes of the VLM.

**Matryoshka Multi-Vector Retrieval (MMR).** With $\mathbf{E}_{\text{meta}}^{(q)} \in \mathbb{R}^{R_q \times D}$ and $\mathbf{E}_{\text{meta}}^{(c)} \in \mathbb{R}^{R_c \times D}$ available, we can compute a late-interaction score between a query $\mathbf{q}$ and a candidate $\mathbf{c}$ as follows:

$$s(\mathbf{q}, \mathbf{c}) = \sum_{i=1}^{R_q} \max_{j \in [1, R_c]} \left\langle \mathbf{E}_{\mathbf{q}}^{(i)}, \mathbf{E}_{\mathbf{c}}^{(j)} \right\rangle. \tag{3}$$

While effective, using all vectors for every instance is not flexible: the index size scales as $O(N \times R_c \times D)$ for $N$ candidates, and the scoring cost scales as $O(R_q \times R_c \times D)$ per pair. We therefore seek a mechanism that maintains strong retrieval quality under tight resources and scales to higher accuracy as more compute is allocated. Inspired by Kusupati et al. (2022), we impose a *prefix-nested* structure on *Meta Embeddings* so that the first few vectors form a coarse summary, and additional vectors refine the representation. Concretely, fix $G$ group sizes for queries so that

$$1 \le r_q^{(1)} < r_q^{(2)} < \cdots < r_q^{(G)} = R_q,$$

and for candidates so that

$$1 \le r_c^{(1)} < r_c^{(2)} < \cdots < r_c^{(G)} = R_c.$$

For any input, define the $g$-th group of query embeddings as $\mathbf{E}^{(q,g)} = \mathbf{E}_{\text{meta}}^{(q)}[1:r_q^{(g)}, :]$, and similarly for candidates $\mathbf{E}^{(c,g)} = \mathbf{E}_{\text{meta}}^{(c)}[1:r_c^{(g)}, :]$. We then compute group-specific late-interaction scores

$$s^{(g)}(\mathbf{q}, \mathbf{c}) = \sum_{i=1}^{r_q^{(g)}} \max_{j \in [1, r_c^{(g)}]} \left\langle \mathbf{E}_{\mathbf{q}}^{(g,i)}, \mathbf{E}_{\mathbf{c}}^{(g,j)} \right\rangle. \tag{4}$$

During training, we optimize contrastive objectives across all groups in parallel, encouraging each prefix to be discriminative on its own while remaining consistent with larger prefixes.

**Training Objective.** Let $\mathcal{B} = \left( \mathbf{q}^{(b)}, \mathbf{c}^{(b)}, \mathbf{c}^{(b,-)} \right)_{b=1}^{B}$ be a minibatch where each query has a corresponding positive candidate $\mathbf{c}^{(b)}$ and one additional hard negative $\mathbf{c}^{(b,-)}$. For each group $g$, we define the similarity scores between query $u$ and candidate $v$ as follows:

$$\mathbf{S}_{u,v}^{(g)} = \tfrac{1}{\tau} s^{(g)}\left( \mathbf{q}^{(u)}, \mathbf{c}^{(v)} \right), \tag{5}$$

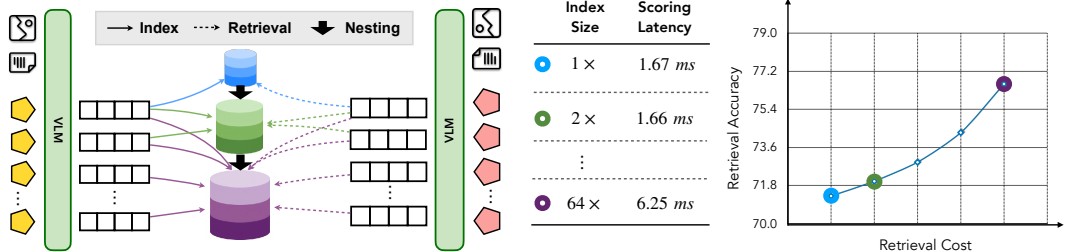

Figure 2: Illustration of test-time scaling with varying retrieval budgets. **Left:** METAEMBED constructs a nested multi-vector index that can be retrieved flexibly given different budgets. **Middle:** How the scoring latency grows with respect to the index size. Scoring latency is reported with 100,000 candidates per query on an A100 GPU. See §5 for full efficiency analysis. **Right:** METAEMBED-7B performance curve with different retrieval budgets. See Figure 3 (b) for full metrics.

with $\tau > 0$ as a temperature hyper-parameter. For query $u$, the denominator of the softmax spans (i) all in-batch candidates $\{\mathbf{c}^{(1)}, \ldots, \mathbf{c}^{(B)}\}$ and (ii) its explicit hard negative $\mathbf{c}^{(u,-)}$. The InfoNCE loss (Oord et al., 2018) for group $g$ is:

$$\mathcal{L}_{\text{NCE}}^{(g)} = -\frac{1}{B} \sum_{u=1}^{B} \log \frac{\exp(\mathbf{S}_{u,u}^{(g)})}{\exp(\mathbf{S}_{u,u}^{(g)}) + \sum_{v \neq u} \exp(\mathbf{S}_{u,v}^{(g)}) + \exp(\frac{1}{\tau} s^{(g)}(\mathbf{q}^{(u)}, \mathbf{c}^{(u,-)}))}. \tag{6}$$

The final loss combines all groups with group-specific hyper-parameters $w_g$ as importance scales:

$$\mathcal{L}_{\text{final}} = \sum_{g=1}^{G} w_g \, \mathcal{L}_{\text{NCE}}^{(g)}. \tag{7}$$

**Test-time Scaling.** The nested design yields a simple accuracy-efficiency knob, as illustrated in Figure 2. At indexing time, one may store only the first $r_c^{(g)}$ vectors for each candidate. At query time, the system selects $(r_q^{(g)}, r_c^{(g)})$ based on latency constraints and computes $s^{(g)}(\mathbf{q}, \mathbf{c})$ for scoring. Coarse prefixes ($g$ small) are ideal for fast retrieval scoring, while larger prefixes ($g$ large) improve precision at the expense of additional compute. Because those groups are optimized in parallel, we can seamlessly adjust the retrieval granularity and budget without retraining the system by selecting a different group size at test-time. In later sections, we refer to the selected combination of group sizes $(r_q^{(g)}, r_c^{(g)})$ as the **retrieval budget**.

## 4 EXPERIMENTS

In this section, we first introduce experimental settings, including models, training data and benchmarks. We then report comprehensive results to showcase the effectiveness and robustness of METAEMBED.

### 4.1 SETTINGS

**Models.** To evaluate the effectiveness of METAEMBED as a training recipe, we conduct experiments on various VLMs of different sizes, including Qwen2.5-VL (Bai et al., 2025), PaliGemma (Beyer et al., 2024) and Llama-3.2-Vision (Grattafiori et al., 2024). Qwen2.5-VL and PaliGemma represent unified multimodal architectures that process text and vision inputs, while Llama-3.2-Vision represents cross-attention-based designs where visual information is integrated into the language model through cross-attention layers. In this section, METAEMBED-3B, -7B and -32B refer to models finetuned on Qwen2.5-VL backbones and METAEMBED-11B is finetuned on the Llama-3.2-Vision model.

**Training.** We train METAEMBED-7B on 32 NVIDIA H100 SXM5 96GB GPUs for 3,500 steps with a global batch size of $2{,}048$, which leads to 30 hours for training. Appendix A.1 has training

Table 1: Precision@1 (%) results on MMEB, which includes 36 tasks across four categories: Classification, Visual Question Answering (VQA), Retrieval, and Visual Grounding. IND and OOD represent the in-domain average and out-of-domain average metrics, respectively. **Bold** denotes the best scores in the subset and the second-best scores are highlighted with underline.

| Models | Size | Per Meta-Task Score | | | | Average Score | | |
|---|---|---|---|---|---|---|---|---|
| | | Classification | VQA | Retrieval | Grounding | IND | OOD | Overall |
| Baseline Models | | | | | | | | |
| CLIP | 428M | 55.2 | 19.7 | 53.2 | 62.2 | 47.6 | 42.8 | 45.4 |
| MagicLens | 613M | 38.8 | 8.3 | 35.4 | 26.0 | – | – | 27.8 |
| UniIR | 428M | 42.1 | 15.0 | 60.1 | 62.2 | – | – | 42.8 |
| ABC | 7B | 60.0 | 31.0 | – | – | – | – | – |
| MM-EMBED | 7B | 48.1 | 32.2 | 63.8 | 57.8 | – | – | 50.0 |
| GME | 7B | 56.9 | 41.2 | 67.8 | 53.4 | – | – | 55.8 |
| VLM2Vec | 7B | 61.2 | 49.9 | 67.4 | 86.1 | 67.5 | 57.1 | 62.9 |
| VLM2Vec-V2 | 2B | 62.9 | 56.3 | 69.5 | 77.3 | – | – | 64.9 |
| MMRet | 7B | 56.0 | 57.4 | 69.9 | 83.6 | 68.0 | 59.1 | 64.1 |
| mmE5 | 11B | 67.6 | 62.7 | 71.0 | 89.7 | 72.4 | 66.6 | 69.8 |
| MoCa-3B | 3B | 59.8 | 62.9 | 70.6 | 88.6 | 72.3 | 61.5 | 67.5 |
| MoCa-7B | 7B | 65.8 | 64.7 | 75.0 | **92.4** | 74.7 | 67.6 | 71.5 |
| B3-7B | 7B | 70.0 | 66.5 | 74.1 | 84.6 | 75.9 | 67.1 | 72.0 |
| METAEMBED – PaliGemma Initialized | | | | | | | | |
| METAEMBED-3B$^{Gemma}$ | 3B | 64.9 | 53.5 | 70.9 | 79.5 | 68.6 | 61.3 | 65.4 |
| METAEMBED – Llama-3.2-Vision Initialized | | | | | | | | |
| METAEMBED-11B | 11B | 66.4 | 42.1 | 74.3 | 91.6 | 65.7 | 64.3 | 65.1 |
| METAEMBED – Qwen2.5-VL Initialized | | | | | | | | |
| METAEMBED-3B | 3B | 62.7 | 68.1 | 71.9 | 79.6 | 73.5 | 63.8 | 69.1 |
| METAEMBED-7B | 7B | 71.3 | 74.2 | 78.7 | 85.4 | 81.8 | 70.0 | 76.6 |
| METAEMBED-32B | 32B | **73.7** | **78.6** | **78.9** | 88.1 | **82.8** | **73.7** | **78.7** |

details for other variants. We use LoRA (Hu et al., 2022) with a rank of 32 and scaling factor $\alpha = 32$ in all training. For models with Matryoshka Multi-Vector Retrieval, we empirically choose $G = 5$ group sizes of $(r_q, r_c)$ as $\{(1,1), (2,4), (4,8), (8,16), (16,64)\}$ and discuss other group size options in §4.3. Group-specific hyper-parameter $w_g$ in Equation 7 is set to 1 following Kusupati et al. (2022). Contrastive training temperature $\tau$ is set to 0.03. We only incorporate MMEB-train (Jiang et al., 2025) and ViDoRe-train (Faysse et al., 2025) with one explicit hard negative from Chen et al. (2025a) for training all variants of METAEMBED.

**Evaluation.** We assess the general multimodal embedding ability of METAEMBED on the Massive Multimodal Embedding Benchmark (MMEB) (Jiang et al., 2024) and use Precision@1 as the evaluation metric. MMEB is an established benchmark covering 36 tasks across four types, including classification, visual question answering (VQA) e.g. ScienceQA (Lu et al., 2022), VizWiz (Gurari et al., 2018), ChartQA (Masry et al., 2022), retrieval across a variety of domains e.g. Visual News (Liu et al., 2021), FashionIQ (Wu et al., 2021), OvenWiki (Hu et al., 2023), and visual grounding e.g. COCO (Lin et al., 2014), RefCOCO (Kazemzadeh et al., 2014; Yu et al., 2016). In addition, to compare with existing multi-vector solutions on text-image retrieval, we evaluate METAEMBED on Visual Document Retrieval Benchmarks (ViDoRe) v2 (Macé et al., 2025) and use average NDCG@5 as the metric. ViDoRe (Faysse et al., 2025) was first introduced to benchmark visual document retrieval capabilities in different domains, and its v2 version mitigates performance saturation by including more generalized settings and incorporating multilingual subsets. We refer to Appendix A.4 for an introduction to selected baseline methods.

## 4.2 MAIN RESULTS

We report the overall multimodal embedding performance of different METAEMBED variants and baseline methods on MMEB in Table 1. Similarly, we present the visual document retrieval performance of METAEMBED and baselines on ViDoRe v2 in Table 2. All METAEMBED results are reported with 16 query-side vectors and 64 candidate-side vectors, i.e. $(r_q, r_c) = (16, 64)$, and we

Table 2: NDCG@5 (%) results on the ViDoRe v2 benchmark, which covers 7 tasks on visual document retrieval. "Syn" denotes synthetic data, "Mul" indicates multilingual tasks, and "Bio" refers to biomedical domains.

| Models | Size | ESG_Human | Eco_Mul | Bio | ESG_Syn | ESG_Syn_Mul | Bio_Mul | Eco | Avg. |
|---|---|---|---|---|---|---|---|---|---|
| Single-Vector Retrieval | | | | | | | | | |
| SigLIP | 652M | 28.8 | 14.0 | 33.8 | 19.8 | 21.9 | 18.2 | 29.8 | 23.8 |
| VLM2Vec | 7B | 33.9 | 42.0 | 38.8 | 36.7 | 38.4 | 29.7 | 51.4 | 38.7 |
| VisRAG-Ret | 3B | 53.7 | 48.7 | 54.8 | 45.9 | 46.4 | 47.7 | 59.6 | 51.0 |
| GME | 7B | **65.8** | 56.2 | 64.0 | 54.3 | 56.7 | 55.1 | 62.9 | 59.3 |
| mmE5 | 11B | 52.8 | 44.3 | 51.3 | 55.1 | 54.7 | 46.8 | 48.6 | 50.5 |
| MoCa-3B | 3B | 63.3 | 57.3 | 62.5 | 58.3 | 54.8 | 59.8 | 62.8 | 59.8 |
| MoCa-7B | 7B | 58.8 | **57.6** | 63.2 | 55.3 | 51.4 | 61.3 | **63.8** | 58.8 |
| Multi-Vector Retrieval | | | | | | | | | |
| ColPali | 3B | 51.1 | 49.9 | 59.7 | 57.0 | 55.7 | 56.5 | 51.6 | 54.5 |
| ColQwen2 | 2B | 62.2 | 53.2 | 61.8 | 53.4 | 54.2 | 56.5 | 61.5 | 57.5 |
| METAEMBED | | | | | | | | | |
| METAEMBED-3B | 3B | 63.7 | 55.5 | 61.7 | 62.6 | 57.4 | 58.7 | 62.3 | 60.3 |
| METAEMBED-7B | 7B | 62.9 | 54.2 | **65.0** | **62.9** | **61.1** | **61.9** | 60.9 | **61.3** |

will discuss the impact of the number of *Meta Embeddings* used in §4.3. We conclude key observations from those metrics as follows.

**METAEMBED delivers substantial improvements over the best existing single-vector baselines at comparable model sizes.** At the 3B scale, METAEMBED achieves 69.1 overall on MMEB, already surpassing MoCa-3B (67.5) with +1.6% relative improvement. At 7B, the margin widens: METAEMBED reaches 76.6, outperforming MoCa-7B (71.5) and mmE5 (69.8) by over 5-7 points. Scaling further to 32B yields 78.7 overall, a clear improvement over both the strongest baselines and our smaller variants. Importantly, the relative gains of METAEMBED increase with model size – while the 3B variant offers competitive results, the 7B and 32B models establish new state-of-the-art performance with the gap over baselines widening as scale increases. This trend suggests that METAEMBED scales more favorably than prior approaches, with benefits compounding in larger regimes.

**The choice of VLM backbone has a pronounced effect on METAEMBED performance across tasks.** METAEMBED-11B with Llama-3.2-Vision backbone shows strong grounding and solid classification abilities, but its VQA score drops sharply to 42.1 – more than 32 points lower than the Qwen2.5-VL-initialized 7B model (74.2). This limitation caps its overall score at 65.1 despite excelling in other subtasks. In contrast, Qwen2.5-VL initialization consistently delivers balanced improvements across all metrics: METAEMBED-7B achieves 76.6, and scaling further to 32B pushes the state of the art to 78.7, with especially strong VQA and retrieval capabilities. We notice that if the underlying base model itself struggles on some domains when used as a generative model, such a weakness directly propagates into METAEMBED as an embedding model. For example, Huang et al. (2025) suggests that Llama-3.2-Vision-11B is less competitive in most zero-shot VQA benchmarks, and such weakness is inherited in METAEMBED-11B.

**METAEMBED demonstrates strong retrieval performance on ViDoRe v2, particularly in multilingual and biomedical domains, despite not being trained on multilingual data.** Even at the 3B scale, METAEMBED matches or surpasses much larger baselines, showing robustness across all seven evaluation tracks. When scaled to 7B, the model yields further gains, with the largest improvements appearing in multilingual and biomedical domains. This is especially noteworthy given that no explicit multilingual data was included during training, suggesting that METAEMBED effectively retains and leverages cross-lingual capabilities from its backbone.

### 4.3 ABLATION STUDIES

To better understand METAEMBED, we design comprehensive ablation studies to investigate its test-time scaling capabilities, the effectiveness of MMR and its robustness across different models.

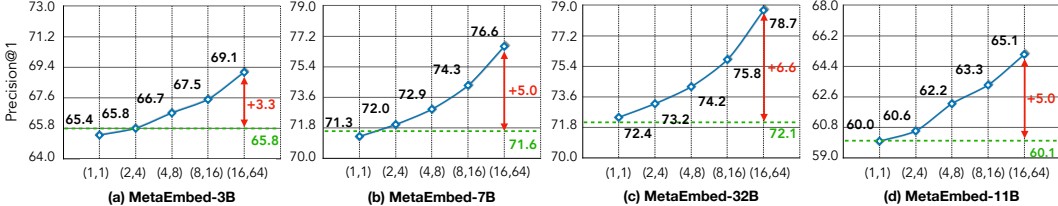

Figure 3: Impact of retrieval budget on MMEB across METAEMBED of varying model sizes. Retrieval budget is denoted as $(r_q, r_c)$, *i.e.* a tuple of the number of *Meta Embeddings* used on query and candidate side. Increasing the retrieval budget from (1,1) to (16,64) consistently improves performance for all model sizes, with larger gains observed in higher-capacity models. The dashed green lines indicate the best single-vector retrieval performance and red arrows indicate the absolute gain (in percentage points) between METAEMBED and single-vector retrieval.

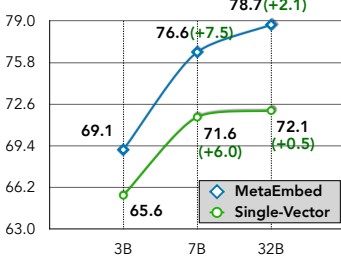

(a) METAEMBED with $(16, 64)$ retrieval budget shows less diminishing returns as model size scales. Green numbers denote the gain compared to the preceding model size.

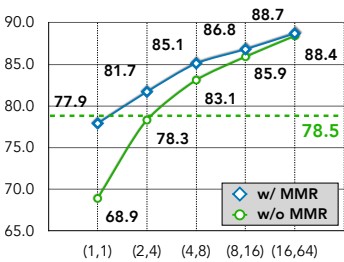

(b) Average NDCG@5 (%) on ViDoRe v1 benchmark with varying retrieval budgets on METAEMBED-3B with and without MMR design.

Figure 4: Ablation studies.

**How does the performance scales with the retrieval cost?** We present the performance plots of different METAEMBED models with respect to retrieval cost in Fig. 3. Data points in each plot correspond to METAEMBED performance with a specific model size with varying test-time retrieval budgets. The dashed green line marks the best-performance single-vector retrieval model with identical training settings. The plots show that across model sizes, the curves rise steadily as more retrieval budget is allocated. While the improvement is modest for smaller models as METAEMBED-3B shows +3.3 points relative gain against the single-vector method, we observe that it becomes more noticeable as the base model size grows and METAEMBED brings the most pronounced improvements on the 32B model with a +6.6 points gain.

**Does METAEMBED apply to pre-trained VLMs of different sizes and architectures?** Fig. 3 (d) already demonstrates that METAEMBED-11B shows advantages when finetuned on different VLM, Llama-3.2-Vision-11B, showing the robustness of our method across architectures. Another key observation is that METAEMBED makes more effective use of larger model capacity compared to single-vector methods. Fig. 4a presents the performance of the two approaches on MMEB under identical training settings as model size increases. We find that METAEMBED achieves more substantial gains than single-vector retrieval. Notably, the improvement of the single-vector baseline from 7B to 32B is no longer statistically significant while METAEMBED still holds a noticeable gain.

**How effective is the MMR design?** To investigate how MMR functions, *i.e.* how it organizes query and candidate information in a nested order of importance, we use average NDCG@5 on ViDoRe-v1 and report test-time scaling curves on two variants of METAEMBED-3B: with and without MMR in Fig. 4b. If MMR is not enabled during training, the flexibility of METAEMBED will be severely constrained, as evidenced by the substantial performance drop under low retrieval budgets. For example, the performance drop hits 9.0 points when using METAEMBED without MMR as a single-vector retrieval model, *i.e.* $(r_q, r_c) = (1, 1)$. Although the gap narrows as retrieval budget increases, the model with MMR consistently outperforms the non-MMR model as shown in the figure. Surprisingly, we find that even at the full budget $(r_q, r_c) = (16, 64)$, the MMR model still performs slightly better, demonstrating that MMR does not sacrifice the original multi-vector retrieval ability at full scale. We additionally report the test-time scaling results on MMEB in Appendix A.2 where MMR shows negligible performance degradation.

Table 3: Efficiency analysis of METAEMBED-7B with different retrieval budgets on an A100 GPU with 100,000 candidates per query with scoring batch size of 1,000. Query encoding and index generation latency are omitted because they remain the same for all variants. Latency refers specifically to scoring latency and mean and standard deviation of latency are reported with 10 runs. Index is stored and compared with `bfloat16` precision (Wang & Kanwar, 2019).

| Retrieval Budget | Scoring FLOPs (G) | Latency (ms) | Index Memory (GiB) | MMEB Acc (%) |
|---|---|---|---|---|
| $(1, 1)$ | 0.71 | $1.67_{\pm 0.13}$ | 0.68 | 71.3 |
| $(2, 4)$ | 5.73 | $1.66_{\pm 0.12}$ | 2.67 | 72.0 |
| $(4, 8)$ | 22.94 | $1.67_{\pm 0.12}$ | 5.34 | 72.9 |
| $(8, 16)$ | 91.75 | $1.92_{\pm 0.12}$ | 10.68 | 74.3 |
| $(16, 64)$ | 733.89 | $6.25_{\pm 0.07}$ | 42.72 | 76.6 |

## 5 DISCUSSION

In this section, we mainly discuss the efficiency of METAEMBED as a flexible multi-vector retrieval method, with a focus on index memory consumption and latency under varying retrieval budgets.

A typical online retrieval process consists of three stages: (a) Query encoding, where the query is processed by the encoder to obtain contextualized embeddings. (b) Scoring, where query embeddings are compared with candidate document embeddings in the index. For single-vector dense retrieval, this is a dot-product operation between pairs of vectors, while in multi-vector retrieval such as METAEMBED it requires late interaction (e.g., MaxSim in Eq. 2) between multiple embeddings. (c) Ranking, a lightweight operation where candidate documents are sorted based on the scores.

We report the efficiency analysis of METAEMBED-7B in Table 3 with the following observations:

1. Although the number of scoring FLOPs grows substantially with larger retrieval budgets, the scoring stage itself is not compute-bounded until the extreme case of $(16, 64)$. The measured latencies remain nearly flat across moderate budgets, demonstrating that GPU throughput can accommodate the additional FLOPs without becoming a bottleneck.

2. The relative contribution of scoring cost to the overall retrieval pipeline is negligible compared to query encoding. For instance, encoding an image query of 1024 tokens requires 42.72 TFLOPs and 788ms. These figures are orders of magnitude larger than the scoring costs reported in Table 3, indicating that efficiency improvements should primarily target encoding rather than scoring with small number of candidates.

3. As a flexible multi-vector retrieval method, index memory consumption can grow proportionally with the retrieval budget. While this can present challenges for large deployments, the issue can be mitigated by using a balanced retrieval budget or more frequent offloading of index data to CPU memory.

Overall, these findings suggest that METAEMBED is efficient in practice. Query encoding dominates latency, scoring is lightweight under most realistic budgets with a small number of candidates, and memory scaling can be controlled by either selecting balanced retrieval budgets or system-level strategies such as CPU swapping.

## 6 CONCLUSION

We present METAEMBED, a new paradigm for multimodal retrieval that rethinks the construction and interaction of embeddings at scale. By leveraging a small set of learnable *Meta Tokens* and training them through our proposed Matryoshka Multi-Vector Retrieval (MMR) framework, METAEMBED organizes information into coarse-to-fine levels of granularity. This design enables flexible late interaction that balances retrieval accuracy, index size, and latency – unlocking test-time scalability for multimodal retrieval. We believe METAEMBED opens a path toward more general, efficient, and controllable multimodal retrieval, bridging the gap between fine-grained expressiveness and large-scale deployability.

## ACKNOWLEDGEMENTS

We thank Anshumali Shrivastava, Xu Han, Norman Huang and Xueyuan Su for insightful discussions and support. V. Ordonez is supported by National Science Foundation CAREER Award No. 2201710 and support from the Ken Kennedy Institute at Rice University.

## ETHICS STATEMENT

The training and evaluation of METAEMBED rely on existing multimodal datasets. While these datasets are widely adopted in the research community, they may contain biases (*e.g.*, cultural, demographic, or linguistic) that can propagate into the learned representations. We acknowledge this risk and emphasize that users should carefully consider dataset composition and potential downstream fairness impacts when deploying such systems.

## REPRODUCIBILITY STATEMENT

Experimental setups (backbones, data, metrics, and budgets) are detailed in §4.2 and §4.3. Main MMEB and ViDoRe results appear in Table 1 and Table 2, while test-time scaling and ablations are summarized in Figures 2, 3, 4a, and 4b. Implementation and compute details (optimizers, LoRA settings, group sizes, temperatures, hardware) are provided in Appendix A.1 (see Table 4); additional ablation tables and comparisons are in Appendix A.2, and baseline descriptions are compiled in Appendix A.4. Efficiency measurements and index sizing used for latency/memory reporting are discussed in §5 and tabulated in Table 3.

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

# A APPENDIX

## A.1 IMPLEMENTATION DETAILS

Table 4: Training details of METAEMBED variants.

|  | Batch Size | Learning Rate | Training Cost | Embedding Dim |
|---|---|---|---|---|
| METAEMBED-3B[Gemma] | 2,048 | $1 \times 10^{-4}$ | 32 H100s for 14h | 2,048 |
| METAEMBED-3B | 2,048 | $1 \times 10^{-4}$ | 32 H100s for 23h | 2,048 |
| METAEMBED-7B | 1,536 | $1 \times 10^{-4}$ | 32 H100s for 30h | 3,584 |
| METAEMBED-11B | 1,024 | $1 \times 10^{-4}$ | 32 H100s for 10h | 4,096 |
| METAEMBED-32B | 1,536 | $1 \times 10^{-5}$ | 64 H100s for 25h | 5,120 |

We list the details of each METAEMBED variant in Table 4. We reserve 1% training data from each subset of MMEB-train as evaluation split and training was early stopped when evaluation loss stops dropping. All models are trained with gradient checkpointing (Chen et al., 2016) to reduce memory usage. We use Adam optimizer (Kingma, 2014) with betas of 0.9, 0.999 and no weight decay and apply a linear learning rate warmup of 500 steps in all training configurations. We use LoRA (Hu et al., 2022) with a rank of 32 and scaling factor $\alpha = 32$ in all training. Contrastive training temperature $\tau$ is set to 0.03. Training was conducted using PyTorch (Paszke et al., 2019) `2.6.0+cu124` and FlashAttention 2.0 (Dao, 2024). For the 3B configuration, we adopt Distributed Data Parallel (Li et al., 2020) while for all other model sizes we use Fully Sharded Data Parallel (FSDP) (Zhao et al., 2023) v2. To prevent distributed hanging when training samples contain no images under FSDP, we pad those samples with a placeholder image to ensure the visual encoder is activated on each GPU. We use the default system prompts of the base models. Each training query in MMEB-train (Jiang et al., 2024) contains a task-specific text instruction, and we keep them during training following common practice in the community. Those instructions are reused during training and evaluation on the same type of tasks.

For efficiency analysis implementation in §5, we store the index in native PyTorch tensors and use PyTorch `torch.einsum` to implement late interaction operation following the practice in Faysse et al. (2025). This controlled setup allows us to compare the cost of different retrieval budgets without additional system-level engineering. In a realistic large-scale deployment, one would typically rely on specialized similarity-search frameworks such as FAISS (Douze et al., 2025) or systems optimized for multi-vector indexes (e.g., WARP (Scheerer et al., 2025), LEANN (Wang et al., 2025)), which provide more efficient memory management and higher throughput. Therefore, the latencies reported in Table 3 should be interpreted as conservative upper bounds rather than results from an optimized production system.

## A.2 DETAILED MMEB ABLATION RESULTS

To further compare the superiority of METAEMBED against both single-vector and multi-vector retrieval methods, we design the following baselines based on the same pre-trained models for fair comparison:

1. single-last: a single-vector retrieval method that uses the last-token hidden state from the last layer as the retrieval representation.
2. single-mean: similar to the above, but applies average pooling over all last-layer hidden states to obtain the retrieval representation.
3. split-$(16, 64)$: a simple multi-vector retrieval baseline where the query-side last-layer hidden states are evenly partitioned into 16 segments, with the mean of each segment taken as 16 query vectors; the same process produces 64 candidate-side vectors. This method does not introduce additional parameters, making it a suitable fixed-length multi-vector retrieval baseline.

Our findings indicate that METAEMBED goes beyond test-time scaling advantages mentioned in the main text: it consistently outperforms the top single-vector method as well as a naive multi-vector baseline. Moreover, the MMR design brings no statistically significant loss, demonstrating that it adds flexibility while maintaining retrieval quality.

Table 5: Comparison between METAEMBED and single-vector & multi-vector retrieval models trained with identical settings. NoMMR indicates Matryoshka Multi-Vector Retrieval (MMR) is disabled. $\Delta$ denotes the difference to the best single-vector retrieval method, *i.e.* single-last.

| | 3B | | 7B | | 32B | | 11B | |
|---|---|---|---|---|---|---|---|---|
| Type | MMEB | $\Delta$ | MMEB | $\Delta$ | MMEB | $\Delta$ | MMEB | $\Delta$ |
| single-last | 65.6 | 0 | 71.6 | 0 | 72.1 | 0 | 60.1 | 0 |
| single-mean | 65.2 | -0.4 | 71.2 | -0.4 | 71.1 | -1.0 | 58.1 | -2.0 |
| split-$(16, 64)$ | 64.2 | -1.4 | 70.1 | -1.5 | 70.5 | -1.6 | 56.0 | -4.1 |
| METAEMBED | | | | | | | | |
| $(1, 1)$ | 65.4 | -0.2 | 71.3 | -0.3 | 72.4 | +0.3 | 60.0 | -0.1 |
| $(2, 4)$ | 65.8 | +0.2 | 72.0 | +0.4 | 73.2 | +1.1 | 60.6 | +0.5 |
| $(4, 8)$ | 66.7 | +1.1 | 72.9 | +1.3 | 74.2 | +2.1 | 62.2 | +2.1 |
| $(8, 16)$ | 67.5 | +1.9 | 74.3 | +2.7 | 75.8 | +3.7 | 63.3 | +3.2 |
| $(16, 64)$ | 69.1 | +3.5 | 76.6 | +5.0 | 78.7 | +6.6 | 65.1 | +5.0 |
| NoMMR-$(16, 64)$ | 69.3 | +3.7 | 77.0 | +5.4 | 79.1 | +7.0 | 66.2 | +6.1 |

## A.3 MMEB PER-TASK SCORE

We present the per-task results of METAEMBED and baseline models on the MMEB benchmark (Jiang et al., 2024) in Table 6. While the in-domain (IND) and out-of-domain (OOD) setup follow the previous multimodal retrieval literatures (Jiang et al., 2025; Chen et al., 2025a; Thirukovalluru et al., 2025; Meng et al., 2025), we believe such an evaluation protocol is subject to known issues arising from the train-test class overlap. As the anonymous reviewer pointed out, ObjectNet (Barbu et al., 2019) and ImageNet (Deng et al., 2009) have 113 overlapped classes. Song et al. (2024) demonstrates that such overlap can bias performance estimates. Therefore, we call for more principled evaluation protocols in the topic of multimodal retrieval.

## A.4 BASELINE METHOD INTRODUCTION

For clarity and completeness, we provide short introductions to the baseline methods considered in Table 1. All performance metrics reported on baseline methods are directly taken from the corresponding original papers or Chen et al. (2025a).

**CLIP (Radford et al., 2021).** CLIP is a dual-encoder trained with contrastive learning on 400M image–text pairs. It learns aligned representations for both modalities, enabling strong zero-shot classification and retrieval capabilities.

**MagicLens (Zhang et al., 2024a).** MagicLens is a lightweight dual-encoder for instruction-guided image retrieval. It is trained in a self-supervised manner on roughly 36.7M (query-image, text instruction, target-image) triplets mined from co-occurring web images.

**UniIR (Wei et al., 2024).** UniIR is a unified, instruction-guided multimodal retriever that handles eight retrieval task formats spanning text, image, and mixed-modality queries/candidates. It is jointly trained on ten heterogeneous datasets, showing robust in-distribution performance and zero-shot generalization across tasks.

**ABC (Schneider et al., 2025).** ABC is an open-source multimodal embedding model designed to unify visual and textual representations under explicit natural language control. To evaluate instruction sensitivity, the authors introduce CtrlBench, a benchmark requiring retrieval conditioned on subtle, interleaved visual and linguistic cues. ABC demonstrates strong generalization on CtrlBench, validating its ability to produce controllable, high-fidelity visual embeddings. It also achieves good performance on MMEB in zero-shot settings.

Table 6: Detailed performance on 36 MMEB tasks. Table style and baseline performance are adopted from Chen et al. (2025a). Rows in yellow indicate metrics of an OOD task.

| Task | CLIP | OpenCLIP | VLM2Vec | MMRet | mmE5 | MoCa-3B | MoCa-7B | METAEMBED-7B |
|---|---|---|---|---|---|---|---|---|
| **Classification (10 tasks)** | | | | | | | | |
| ImageNet-1K | 55.8 | 63.5 | 74.5 | 58.8 | 77.6 | 75.4 | 78.0 | 88.1 |
| N24News | 34.7 | 38.6 | 80.3 | 71.3 | 82.1 | 80.9 | 81.5 | 84.3 |
| HatefulMemes | 51.1 | 51.7 | 67.9 | 53.7 | 64.3 | 70.6 | 77.6 | 78.7 |
| VOC2007 | 50.7 | 52.4 | 91.5 | 85.0 | 91.0 | 87.0 | 90.0 | 94.1 |
| SUN397 | 43.4 | 68.8 | 75.8 | 70.0 | 77.9 | 74.8 | 76.8 | 83.3 |
| Place365 | 28.5 | 37.8 | 44.0 | 43.0 | 42.6 | 38.8 | 43.0 | 48.6 |
| ImageNet-A | 25.5 | 14.2 | 43.6 | 36.1 | 56.7 | 39.7 | 52.7 | 60.9 |
| ImageNet-R | 75.6 | 83.0 | 79.8 | 71.6 | 86.3 | 75.4 | 83.0 | 88.1 |
| ObjectNet | 43.4 | 51.4 | 39.6 | 55.8 | 62.2 | 31.3 | 45.2 | 56.8 |
| Country-211 | 19.2 | 16.8 | 14.7 | 14.7 | 34.8 | 24.0 | 30.4 | 35.4 |
| *All Classification* | 42.8 | 47.8 | 61.2 | 56.0 | 67.6 | 59.8 | 65.8 | 71.8 |
| **VQA (10 tasks)** | | | | | | | | |
| OK-VQA | 7.5 | 11.5 | 69.0 | 73.3 | 67.9 | 40.0 | 36.9 | 76.7 |
| A-OKVQA | 3.8 | 3.3 | 54.4 | 56.7 | 56.4 | 54.6 | 57.1 | 69.1 |
| DocVQA | 4.0 | 5.3 | 52.0 | 78.5 | 90.3 | 93.0 | 94.3 | 96.5 |
| InfographicsVQA | 4.6 | 4.6 | 30.7 | 39.3 | 56.2 | 67.7 | 77.2 | 82.1 |
| ChartQA | 1.4 | 1.5 | 34.8 | 41.7 | 50.3 | 64.1 | 69.8 | 78.3 |
| Visual7W | 4.0 | 2.6 | 49.8 | 49.5 | 51.9 | 61.6 | 58.5 | 72.9 |
| ScienceQA | 9.4 | 10.2 | 42.1 | 45.2 | 55.7 | 45.4 | 59.2 | 57.4 |
| VizWiz | 8.2 | 6.6 | 43.0 | 51.7 | 52.8 | 52.3 | 46.2 | 55.5 |
| GQA | 41.3 | 52.5 | 61.2 | 59.0 | 62.1 | 66.9 | 71.6 | 67.2 |
| TextVQA | 7.0 | 10.9 | 62.0 | 79.0 | 83.5 | 83.1 | 75.8 | 85.8 |
| *Avg. VQA* | 9.1 | 10.9 | 49.9 | 57.4 | 62.7 | 62.9 | 64.7 | 74.1 |
| **Retrieval (12 tasks)** | | | | | | | | |
| VisDial | 30.7 | 25.4 | 80.9 | 83.0 | 73.7 | 80.5 | 84.5 | 87.4 |
| CIRR | 12.6 | 15.4 | 49.9 | 61.4 | 54.9 | 55.7 | 53.4 | 65.9 |
| VisualNews_t2i | 78.9 | 74.0 | 75.4 | 74.2 | 77.7 | 74.4 | 78.2 | 80.6 |
| VisualNews_i2t | 79.6 | 78.0 | 80.0 | 78.1 | 83.4 | 77.8 | 83.1 | 84.3 |
| MSCOCO_t2i | 59.5 | 63.6 | 75.7 | 78.6 | 76.2 | 76.4 | 79.8 | 85.2 |
| MSCOCO_i2t | 57.7 | 62.1 | 73.1 | 72.4 | 73.6 | 72.6 | 73.9 | 82.2 |
| NIGHTS | 60.4 | 66.1 | 65.5 | 68.3 | 68.8 | 67.4 | 66.7 | 75.5 |
| WebQA | 67.5 | 62.1 | 87.6 | 90.2 | 88.1 | 90.6 | 91.4 | 93.0 |
| FashionIQ | 11.4 | 13.8 | 16.2 | 54.9 | 28.6 | 22.2 | 28.9 | 34.1 |
| Wiki-SS-NQ | 55.0 | 44.6 | 60.2 | 24.9 | 65.2 | 73.3 | 82.7 | 83.6 |
| OVEN | 41.1 | 45.0 | 56.5 | 87.5 | 77.3 | 75.9 | 80.4 | 83.3 |
| EDIS | 81.0 | 77.5 | 87.8 | 65.6 | 83.6 | 80.8 | 96.9 | 89.1 |
| *Avg. Retrieval* | 53.0 | 52.3 | 67.4 | 69.9 | 71.0 | 70.6 | 75.0 | 78.6 |
| **Visual Grounding (4 tasks)** | | | | | | | | |
| MSCOCO | 33.8 | 34.5 | 80.6 | 76.8 | 85.0 | 80.2 | 84.6 | 78.0 |
| RefCOCO | 56.9 | 54.2 | 88.7 | 89.8 | 92.7 | 92.1 | 94.0 | 94.7 |
| RefCOCO-matching | 61.3 | 68.3 | 84.0 | 90.6 | 88.9 | 92.8 | 95.5 | 93.4 |
| Visual7W-pointing | 55.1 | 56.3 | 90.9 | 77.0 | 92.3 | 89.5 | 95.3 | 87.2 |
| *Avg. Grounding* | 51.8 | 53.3 | 86.1 | 83.6 | 89.7 | 88.7 | 92.4 | 88.3 |
| **Final Score (36 tasks)** | | | | | | | | |
| *All IND Avg.* | 37.1 | 39.3 | 67.5 | 59.1 | 72.4 | 72.3 | 74.7 | 81.8 |
| *All OOD Avg.* | 38.7 | 40.2 | 57.1 | 68.0 | 66.6 | 61.5 | 67.6 | 71.0 |
| *All Tasks Avg.* | 37.8 | 39.7 | 62.9 | 64.1 | 69.8 | 67.5 | 71.5 | 76.6 |

**MM-Embed (Lin et al., 2024).** MM-Embed converts a multimodal large language model into a universal bi-encoder for retrieval. It is fine-tuned with modality-aware hard negatives across diverse retrieval datasets to improve cross-modal alignment.

**GME (Zhang et al., 2024b).** The General Multimodal Embedder is trained on a large synthetic dataset containing diverse multimodal queries and documents. It introduces fused-modal training examples (mixed text–image inputs) to enable universal any-to-any modality retrieval.

**VLM2Vec (Jiang et al., 2025).** VLM2Vec transforms a pretrained vision-language model into a universal embedding model through instruction-tuned contrastive learning. It is trained on the Massive Multimodal Embedding Benchmark (MMEB), covering 36 tasks across classification, VQA, retrieval, and grounding.

**MMRet (Zhou et al., 2024).** MMRet builds on the MM-Embed framework but introduces further refinements in negative sampling and large-scale fine-tuning on massive synthetic dataset, making it one of the strongest retrieval-centric embedding models in prior work.

**mmE5 (Chen et al., 2025b).** mmE5 extends the multilingual text embedding model E5 into the multimodal setting. It is trained with multilingual and multimodal signals, leveraging synthetic image-text pairs and hard negatives, and achieves strong state-of-the-art results on MMEB prior to more recent models.

**MoCa (Chen et al., 2025a).** MoCa (Modality-aware Causal Pre-training) introduces a two-stage process: modality-aware continual pre-training to adapt causal vision–language models for bidirectional encoding, followed by heterogeneous contrastive fine-tuning across text, image, and mixed modality pairs. We evaluate both MoCa-3B and MoCa-7B, which show competitive overall performance among baselines on MMEB.

**B3 (Thirukovalluru et al., 2025)** B3-7B is a 7B-parameter instruction-tuned multimodal retriever with specialized batch mining techniques. It achieves strong results on MMEB and serves as an additional competitive baseline.

## A.5 THE USE OF LARGE LANGUAGE MODELS (LLMS)

We used large language models (LLMs) solely for minor language editing purposes, such as catching grammar errors and improving clarity of expression. The LLM did not contribute to research ideation, analysis, or substantive writing of the paper. All conceptual development, methodology, results, and interpretations were conducted entirely by the authors.

