# OpenReview forum: "MetaEmbed: Scaling Multimodal Retrieval at Test-Time with Flexible Late Interaction"
_ICLR.cc/2026/Conference — ICLR 2026 Oral_

### Official Review · Reviewer_rgEq · 2025-10-23

**Soundness:** 4
**Presentation:** 4
**Contribution:** 2
**Rating:** 6
**Confidence:** 4

**Summary:**

This paper introduces METAEMBED, a multimodal embedding learning framework designed for scalable late interaction. METAEMBED adds a small number of learnable Meta Tokens to multimodal inputs and trains them using a Matryoshka Multi-Vector Retrieval (MMR) strategy, which organizes embeddings hierarchically to enable test-time scaling. Specifically, MMR enables test-time scaling balancing retrieval accuracy and latency.
The authors evaluate METAEMBED on MMEB and ViDoRe v2 benchmarks across multiple vision-language backbones (e.g., Qwen2.5-VL, PaliGemma, and LLaMA-3.2-Vision), showing state-of-the-art retrieval performance and flexible inference efficiency across different VLM varying backbones and model sizes.

**Strengths:**

1. Strong performance: METAEMBED demonstrates strong performance across diverse domains and setups over baseline retrievers.
2. General applicability across VLM backbones and model sizes: METAEMBED is evaluated on Qwen2.5-VL, PaliGemma, and LLaMA-3.2-Vision, show that it generalizes to different model architectures and parameter scales.
3. Comprehensive analysis of METAEMBED: the authors provide comprehensive ablations such as the effectiveness of MMR component, efficiency analysis, analysis on different VLM backbones.

**Weaknesses:**

1. Limited originality beyond adaptation of MRL: The main innovation lies in adapting Matryoshka Representation Learning (MRL) to multi-vector retrieval. While this adaptation is meaningful, much of the practical foundation (i.e., multimodal representation learning with Matryoshka nesting) derives from prior work (Cai et al. 2025).
2. Insufficient analysis of efficiency in training: Although efficiency is discussed in the context of query encoding at test-time (Table 3). Comparing computational costs in training across different retriever methods can further strengthen the paper, as additional training costs incurred could be a trade-off for enabling flexible test-time compute.
3. Empirical gaps in hyperparameter justification: the paper fixes the group sizes to be G=5 without ablation on group size for MRL. A sensitivity study on different group configurations would strengthen the argument for generality.

**Questions:**

1. In Table 2, ColQwen2 should correspond to a 3B model rather than 2B.
2. “P” in L182 likely refers to number of visual patch tokens and should be denoted.
3. How do you empirically choose a group size of 5 and with these configurations for MRL?

---

> ### Author Response · Authors · 2025-11-18
>
> We are grateful for your positive recognition of MetaEmbed and would love to address your concerns below.
>
> ---
>
> > W1. Limited Originality
>
> Thank you for raising this concern. Our work is *inspired* by the general Matryoshka idea as indicated in Lines 86 & 197, however **we did not** derive **from Cai et al.**
>
> Cai et al. (Matryoshka Multimodal Models, or “M3”) focus on large multimodal *generative* models, where nested visual tokens are used to **dynamically allocate visual token budgets** when feeding images into an LLM for **generation**. In contrast, MetaEmbed **is an** **embedding** **and** **retrieval** **framework** built around multi-vector late interaction.
>
> Conceptually, both works share the high-level idea of organizing information in a coarse-to-fine, Matryoshka-style structure, but the concrete problems, architectures, and training mechanisms differ substantially. M3 does not address late interaction, multi-vector retrieval, or test-time control of index granularity, and its method is not directly applicable to large-scale multimodal retrieval systems.
>
> We hope this can clarify the originality of our work.
>
> ---
>
> > W2. Comparison w/ baselines in terms of training costs.
>
> Good point! We believe training cost is also an important dimension to evaluate a multimodal retrieval system. Unfortunately, most multimodal retrieval papers do not report training GPU hours as we do, therefore it is very hard to directly compare the training cost. In the next version, we will add a qualitative comparison table regarding training costs upon the consultation of the authors of those baseline methods if they are willing to share the costs.
>
> In addition to training GPU hours, we can also measure the training costs by looking at the training data volume. For example, one of the strongest baselines MoCA [1] introduces a modality-aware continual pre-training stage on roughly 30B tokens on 32 H100 GPUs, followed by a separate heterogeneous contrastive learning stage on millions of multimodal pairs including MMEB and mmE5 data. In contrast, our method **does not introduce any extra long-stage pre-training** & synthetic data beyond the standard retriever training pipeline.
>
> ---
>
> > W3 & Q3. How do you empirically choose a group size of 5 and with these configurations for MRL? A sensitivity study on different group configurations would strengthen the argument for generality.
>
> Our choice of group sizes {(1, 1), (2, 4), (4, 8), (8, 16), (16,64)} is guided by two simple principles.
>
> First, on the query side we follow the original MRL [2] intuition of **exponentially** increasing the effective representation size with the budget. In other words, we roughly double capacity when moving to a larger group.
>
> Second, on the candidate side we allocate more vectors than on the query side, as candidates typically carry richer information and benefit from additional capacity.
>
> Due to the short rebuttal period, we did not run an exhaustive search over all possible groupings. However, **in our preliminary experiments we find that the performance is not very sensitive to moderate changes in the group settings**. For instance, an additional large group (32, 128) did not bring further improvement on the validation set while incurring higher training and memory costs.
>
> ---
>
> > Q1 & Q2. ColQwen2 is a 2B model. $P$ denotes the number of visual tokens.
>
> Thank you for catching these issues! We already revised the submission and highlighted the difference in the updated submission.
>
> ---
>
> We sincerely thank you again for your efforts in reviewing which helped a lot in improving our paper. We hope our responses above could address your concern and please feel free to leave any comments if you have.
>
> References:
>
> [1] Chen, Haonan, et al. "MoCa: Modality-aware Continual Pre-training Makes Better Bidirectional Multimodal Embeddings." *arXiv preprint arXiv:2506.23115* (2025).
>
> [2] Kusupati, Aditya, et al. "Matryoshka representation learning." *Advances in Neural Information Processing Systems* 35 (2022): 30233-30249.

---

### Official Review · Reviewer_g6en · 2025-10-30

**Soundness:** 3
**Presentation:** 4
**Contribution:** 3
**Rating:** 8
**Confidence:** 5

**Summary:**

The paper introduces METAEMBED, a novel framework designed to enhance multimodal retrieval by addressing the key challenges of scalability, efficiency, and fine-grained retrieval accuracy. Existing methods either compress queries and candidates into a single vector, losing fine-grained information, or generate too many vectors, leading to computational inefficiencies. METAEMBED introduces Meta Tokens, which are small, learnable tokens appended to both queries and candidates, and their final contextualized representations serve as compact yet expressive multi-vector embeddings during retrieval. By employing a Matryoshka Multi-Vector Retrieval (MMR) mechanism, the model organizes embeddings into hierarchical groups, enabling flexible scaling at test-time where users can adjust retrieval quality against computational resources.

**Strengths:**

1. The introduction of Meta Tokens and the Matryoshka Multi-Vector Retrieval (MMR) approach provides a unique solution to the scalability and efficiency issues in multimodal retrieval, allowing flexible scaling at test-time.
2. METAEMBED demonstrates remarkable scalability, allowing users to adjust retrieval quality based on computational constraints without retraining the model, a critical advancement for real-world deployment.
3. The paper provides extensive evaluations across various model sizes and VLM architectures, showcasing METAEMBED's robustness and effectiveness in diverse scenarios, including challenging tasks like visual document retrieval and multimodal embedding.

**Weaknesses:**

1. The design intuition of meta embedding is somewhat similar to the concept of query extension in the early information retrieval field. Could the authors further differentiate between these two design intuitions?
2. The approach of using embedding with different granularities for retrieval is promising. Have the authors explored or statistically analyzed which granularity of embedding provides better discrimination? This could offer valuable guidance for future embedding extraction.

**Questions:**

Please refer to the weaknesses.

---

> ### Author Response · Authors · 2025-11-18
>
> Thank you for the positive feedback and encouraging comments. We appreciate your recognition of the uniqueness and scalability of MetaEmbed. Below, we provide detailed response to your comments:
>
> ---
>
> > W1. Difference between MetaEmbed and the concept of query expansion.
>
> We thank the reviewer for drawing this connection. While our intuition of using extra capacity to better represent a query is similar to the early query expansion, MetaEmbed differs from classical query expansion in two ways:
>
> 1. Classific query expansion **operates in the language / token space**. It adds or re-weights extra words, such as synonyms, related terms into the query text, but the retrieval model still scores over the expanded query set using the same underlying representation. MetaEmbed **operates in the** **embedding space**. We append a fixed set of learnable **Meta Tokens** to the input sequence and use their *final hidden states* as the actual multi-vector embedding used in late interaction. Such design could be seen as a soft, differentiable query expansion or re-weighting mechanism.
> 2. MetaEmbed steps **beyond query expansion**. In multimodal retrieval, the candidate (page/image) usually carries much richer information than the query, so we also assign Meta Tokens on the candidate side. This is different from classical **query expansion design which only focuses on improving the query quality.**
>
> ---
>
> > W2. Which granularity of embedding provides better discrimination?
>
> In our multi-vector retrieval setting, we do not see one **best** granularity in isolation. Instead, different Meta Embeddings usually play complementary roles, and using more vectors generally improves retrieval. That said, if the question is which single vector is the most discriminative, our analysis shows it is the first Meta Embedding. We study this in revised Appendix A.4 using a visualization mechanism inspired by ColPali. Given a text query, we compute the similarity between each query-side Meta Embedding and the last-layer image patch representations of a candidate image, and overlay these similarities as heatmaps on the image. This lets us directly inspect which regions each Meta Token focuses on. We find that the **1st Meta Token consistently concentrates its similarity on the key image patches that are semantically aligned with the query**.
>
> In conclusion, we believe the first few embeddings capture the most salient information, while subsequent embeddings refine and enrich the representation. Thus, in terms of *per-vector discrimination*, the 1st Meta Embedding is the most discriminative, but the **combination** of multiple Meta Embeddings yields the best overall retrieval performance.
>
> ---
>
> Thanks again for your review to make MetaEmbed into a better shape. We hope our responses above could address your concerns. We are always happy to engage in discussions.

---

### Official Review · Reviewer_dVyq · 2025-10-31

**Soundness:** 4
**Presentation:** 3
**Contribution:** 4
**Rating:** 8
**Confidence:** 4

**Summary:**

This paper introduces a framework designed to resolve the critical trade-off between expressiveness and efficiency in multimodal retrieval by using a small set of learnable "Meta Tokens" to form a compact multi-vector representation. The central contribution is the Matryoshka Multi-Vector Retrieval (MMR) training objective, which organizes information into a coarse-to-fine structure. This enables a single model to be flexibly scaled at test-time to meet varying accuracy and efficiency demands without retraining.
Overall, this is a quality paper with interesting findings that could have significant applications for the research community. Although some questions and concerns remain, as detailed below, I am willing to raise my score if these points are adequately addressed.

**Strengths:**

- Novel Combination of Existing Concepts: novel synthesis of learnable tokens, late-interaction, and Matryoshka learning, resulting in the technically solid Matryoshka Multi-Vector Retrieval objective.

- The experimental evaluation is thorough and convincing, demonstrating state-of-the-art performance on diverse benchmarks like MMEB and ViDoRe v2.

**Weaknesses:**

- Lack of Interpretability and Qualitative Analysis: The "Meta Tokens" are treated as a black box. The paper offers no analysis or visualization to explain what these tokens learn, missing an opportunity to provide deeper scientific insight beyond a strong engineering result.

- A key motivation presented in the introduction of the paper is solving the efficiency problem for multimodal-to-multimodal (universal) retrieval, yet no experiments on such a task are presented. This leaves a core claim of the paper unsubstantiated by empirical evidence. Specifically, as far as I know, the evaluation on the MMEB and ViDoRe v2 benchmarks lacks an image-to-image retrieval task (which is different from the visual document retrieval task or image+text retrieval task), which would be a key area for qualitative analysis.

**Questions:**

- (Regarding the 2nd weakness) To better understand the mechanism behind METAEMBED, could you provide some qualitative analysis of the learned Meta Embeddings? For instance, what kinds of visual or textual concepts do different tokens appear to specialize in? Does the first token of an MMR-trained model consistently capture more global, summary-level information compared to later tokens, which might focus on finer details?

-The choice of MMR group sizes {(1, 1), (2, 4), (4, 8), (8, 16), (16,64)} appears somewhat heuristic. How sensitive is the model's performance to this specific configuration? For example, what happens if a more linear scaling is used, and what was the process for selecting this particular set of nested budgets?

-The split-(16,64) baseline in Table 5 performs worse than the single-last baseline. This is counter-intuitive, as one might expect a multi-vector representation, even a naive one, to retain more information. Does this imply that without explicit training tailored for late interaction, simply partitioning a representation is detrimental?

---

> ### Author Response · Authors · 2025-11-18
>
> Thank you for the positive comments! We are happy to see the convincing experiments and novelty earn your appreciation and confidence in our work. We will address your concerns below.
>
> ---
>
> >  W1 & W2 & Q1. Interpretability and Qualitative Analysis
>
> We thank the reviewer for raising the concern about interpretability. We agree that understanding what the meta tokens learn is important, and we have now added a qualitative analysis in **Appendix A.4** to address this.
>
> Specifically, we introduce a visualization mechanism from ColPali. Given a query text, we compute the similarity between each **query-side Meta Embedding** and the last-layer representations of image patches in the candidate image. We then visualize this as a heatmap over the image. This allows us to directly inspect which regions each Meta Token focuses on. Our analysis reveals that in MetaEmbed MMR enabled, the **1st Meta Token** consistently concentrates its similarity on key image patches that are semantically aligned with the query, while **later Meta Tokens** exhibit a more diffuse similarity pattern that refines and complements the initial focus. In contrast, in the model **without MMR**, different query tokens show very similar similarity maps and do not specialize or differentiate in what they capture. This behavior exactly echoes the intended design of the MMR mechanism and Meta Tokens, where the first few embeddings capture the most salient information and subsequent embeddings refine the overall representation.
>
> ---
>
> > Q2. What was the process for selecting this particular set of nested budgets? Why {(1, 1), (2, 4), (4, 8), (8, 16), (16,64)}?
>
> The specific group size, {(1, 1), (2, 4), (4, 8), (8, 16), (16,64)}, is chosen by the following design principles:
>
> 1. Exponential growth on the query side: we follow the original MRL idea of doubling the effective representation size as we move to a larger budget. This gives us a small number of groups that still cover a wide range of budgets.
> 2. More capacity on the candidate side: we allocate more embeddings to the candidate side to better capture its richer content.
>
> We acknowledge that such choice is indeed heuristic, but our experiments suggest that performance is not sensitive to modest changes in this configuration. Due to the short rebuttal period, we did **not** perform an exhaustive search over all possible schedules, such as full linear scaling. And we do not claim our choice is optimal. However, in preliminary studies where we extended the schedule with an additional large group (32, 128), we observed no further improvement on the validation set, while incurring extra training and memory cost. This suggests that beyond a certain point, **increasing the number of vectors per group gives diminishing returns under our setup.**
>
> ---
>
> > Q3. Why split-(16,64) baseline in Table 5 perform worse than the single-last baseline?
>
> Good point! While we also agree that the result is counter-intuitive at first glance, we think split-(16,64) has some inherent limitations. First, **it does not introduce any additional learnable** parameters for compressing last-layer representations into multi-vector, so its expressiveness is inherently limited.
>
> Moreover, the split operation is not adaptive to the sequence length. For example, when the query is short (e.g., a single token), split-(16,64) degenerates to a (1,64) case and behaves essentially like the single-to-multi baseline. Second, even for longer sequences, the hard partition may cut across meaningful semantic units, so the resulting vectors do not correspond to coherent, interaction-friendly tokens in the late-interaction sense. And **the model has to figure out a way to adapt to the length-variant multi-vector retrieval process.**
>
> In summary, these observations suggest that directly partitioning last-layer VLM representations, without training explicitly for late interaction, is not an effective way to construct multi-vector representations and can indeed be detrimental.
>
> ---
>
> We truly appreciate your time and constructive feedback. Should you have any additional questions or suggestions, we would be happy to provide further clarification.

---

### Official Review · Reviewer_L8o6 · 2025-10-31

**Soundness:** 3
**Presentation:** 3
**Contribution:** 2
**Rating:** 6
**Confidence:** 4

**Summary:**

The authors propose MetaEmbed which uses grouped multimvector embeddings to scale the "test-time compute" used during retrieval. MetaEmbed introduces a tradeoff between resources (computation, memory), and multimodal retrieval quality. The approach is validated on both general multimodal retrieval tasks (MMEB) and document retrieval (ViDoRe).

**Strengths:**

- The manuscript is clearly written and well structured.

- The introduction of a latency-accuracy tradeoff mechanism into multimodal retrieval is novel and well-motivated.

- The authors demonstrate the value of MetaEmbed across a good variety of multimodal embeddings tasks (both general and visual documents).

- The MMEB and ViDoRe results are mostly fair, with the majority of compared works having similar training distributions.

**Weaknesses:**

- The training data has large distribution overlap with evaluation tasks (especially on key tasks, like ImageNet-1K, ViDORe). Therefore, it is unclear how well the model will generalize.

- It is unclear in tables 1 and 2 which models are trained on MMEB/ViDoRe (and therefore, strongly advantaged). For clarity, it would be valuable to specify which models are in-distribution vs out-of-distribution when reporting performance.

- Several relevant training settings are not included in appendix A.1, such as Optimizer, optimizer hyperparameters, weight decay, learning rate scheduling, was temperature fixed [1] or trained [2]?

- The analysis of latency is limited and potentially misleading. How is the index stored? A realistic setup would use an appropriate framework like FAISS [3]. Can the authors provide more details about their inference setup (table 3) and justify its relationship to a realistic deployment?

- A common use-case would be short text queries and a much larger candidate pool, in which case scoring latency may become significant.

- The device memory requirements for the best performing group settings are very significant, even for the small candidate pool size. This is both a constraint in itself, and can incur significant performance penalties if the index cannot fit into GPU memory.

Minor:

- The reproducibility statement says: “Implementation and compute details (optimizers, LoRA settings, group sizes, temperatures, hardware) are provided in Appendix A.1.” However, some of these details are found elsewhere, whereas others seem to be missing entirely. Ideally, these settings should all be in A.1, even if it results in some duplication from the main text.

- Potentially relevant related work: [4] seems to corroborate the importance of VLM backbone choice for multimodal retrieval, and may be worth including in section 4.2.

**Questions:**

Some questions can be found in the section above, in addition:

- Can the authors provide a per-task breakdown of MMEB? Although averages over MMEB tasks are often reported, not all tasks in MMEB are equally useful when evaluating a model, therefore a per-task breakdown would be useful.

- How are the text queries templated into context? Are task-specific system prompts reused between training and evaluation?

- A comparison between the late interaction approach and a single vector that controls for dimension would be interesting (potentially created by concatenating across the sequence dimension).

[1] R. Meng et al., “VLM2Vec-V2: Advancing Multimodal Embedding for Videos, Images, and Visual Documents,” 2025, doi: 10.48550/arxiv.2507.04590.
[2] C. Jia et al., “Scaling Up Visual and Vision-Language Representation Learning With Noisy Text Supervision,” 2021, doi: 10.48550/arxiv.2102.05918.
[3] M. Douze et al., “THE FAISS LIBRARY,” IEEE transactions on big data, pp. 1–17, 2025, doi: 10.1109/TBDATA.2025.3618474.
[4] B. Schneider, F. Kerschbaum, and W. Chen, “ABC: Achieving Better Control of Visual Embeddings using VLLMs,” Transactions on Machine Learning Research, vol. 2025-, 2025.

---

> ### Author Response · Authors · 2025-11-18
> **Response (Parts 1/2)**
>
> Thank you for your thoughtful comments and positive feedback on MetaEmbed! We are glad that you find MetaEmbed novel and well-motivated. We would love to address your questions point by point below.
>
> ---
>
> > W1. Unclear generalization ability
>
> We agree that simply training and evaluating on closely related datasets can obscure how well a model truly generalizes. In our case, however, we believe the evidence in the paper supports generalization beyond the training distribution.
>
> Our training data is deliberately restricted to the **training splits** of MMEB-train and ViDoRe-train, and MMEB-eval explicitly separates **in-domain (IND)** and **out-of-domain (OOD)** tasks, where the OOD subset contains tasks whose data sources are not included in MMEB-train. Under this stricter OOD protocol, MetaEmbed-7B achieves an average **OOD score of 70.0**, compared to **67.6 for MoCa-7B** and **67.1 for B3-7B**.
>
> In addition, we do not introduce explicit multilingual data during training, yet MetaEmbed achieves strong performance, particularly in the multilingual and biomedical tracks of ViDoRe v2 as indicated in Lines 367-373. This suggests that MetaEmbed can leverage the backbone’s capabilities to generalize to new domains and languages **that are not directly covered by the training data.**
>
> ---
>
> > W2. Clarify the training data of models reported in Tables 1 and 2.
>
> We thank the reviewer for pointing out this ambiguity. All models reported in Tables 1 and 2 are trained on the same mixture of MMEB-train and ViDoRe-train data as indicated in Lines 303-305, and the tables report results for models trained on this mixture.
>
> ---
>
> > W3 & Minor W1. Missing training configuration in Appendix A.1
>
> Thanks for the catch! In the rebuttal revision PDF, we add the detailed and complete training configuration as requested and highlight the difference in Appendix A.1. Specifically, we use Adam optimizer with betas of 0.9, 0.999 and no weight decay and apply a linear learning rate warmup of 500 steps in all training configurations. Contrastive training temperature $\tau$ is set to $0.03$ and remains frozen during training.
>
> ---
>
> > W4. Details of inference setup in Table 3. Justification of the setup of a realistic deployment.
>
> We thank the reviewer for the question about our latency analysis.
>
> For the efficiency analysis in Table 3, we **store the document index as native PyTorch tensors** and implement the late interaction via `torch.einsum`, following the implementation practice of ColPali. This setup allows us to measure the *relative* cost of different retrieval budgets in a controlled environment, without additional engineering optimizations.
>
> We agree that a realistic large-scale deployment would use a specialized framework such as **FAISS** [1] for vector search, or even systems optimized specifically for multi-vector indexes such as **WARP** [2]. Such frameworks typically provide both better memory layouts and higher throughput. Our current numbers should therefore be viewed as a **conservative upper bound** on latency rather than an optimized production deployment.
>
> **We revised the Appendix A.1** to (1) explicitly describe the index storage (PyTorch tensors) and scoring implementation, (2) clarify that Table 3 is based on this reference-style setup, and (3) add a short discussion that in production one would use FAISS or multi-vector-optimized systems.
>
> ---
>
> > W5 & W6. A large candidate pool might make scoring latency and memory consumption become significant.
>
> We agree that in scenarios with short queries and a very large candidate pool, scoring latency & large memory usage are two important practical concerns. However, this issue is not specific to our method. It is shared by essentially all multi-vector retrieval approaches that move beyond single-vector dot-product scoring. In our setting, candidate representations are precomputed offline, and at query time the cost scales with the number of Meta Embeddings per candidate, which we already keep modest.
>
> From a systems perspective, there are several well-established techniques to mitigate latency & memory consumption, such as quantization, approximate nearest neighbor (ANN) search over sub-vectors, and compressed indexes. For example, LEANN [3] uses low-storage vector indices that reduce memory footprint while retaining search efficiency.
>
> Incorporating such indexing and compression strategies into MetaEmbed is largely orthogonal to the core contribution of our paper and thus outside the current scope, but we view index efficiency for large candidate pools as an important direction for future work. **We clarified this limitation and discussed these engineering remedies in the Appendix A.1.**
>
> (To be continued)

---

> ### Author Response · Authors · 2025-11-18
> **Response (Part 2/2)**
>
> > Minor W2. Potential related work
>
> We thank the reviewer for suggesting the missing related work. This work studies how to use a vision-language model backbone to integrate image features with natural language instructions, which is highly related to MetaEmbed design. We have added it to Section 4.2 and discuss it in Appendix A.4.
>
> ---
>
> > Q1. MMEB sub-task scores.
>
> The per-task scores are now presented in Table 6.
>
> ---
>
> > Q2. Task-specific prompt setup.
>
> We now clarified the prompt setup in Appendix A.1. Specifically, we use the default system prompts of the base models. Each training query in MMEB-train contains a task-specific text instruction, and we keep them during training following common practice in the community. Those instructions are reused during training and evaluation on the same type of tasks.
>
> ---
>
> > Q3. Single-vector concat v.s. Multi-vector late interaction
>
> Good idea! We agree that controlling for dimensionality is important, and we did attempt the suggested baseline where we concatenate Meta Embeddings across the sequence dimension to form a single high-dimensional vector. In the case of MetaEmbed-7B, it results in a 3584*16-dim vector.
>
> However, this setup proved extremely difficult to optimize in practice. After L2 normalization, the effective per-coordinate magnitude became very small, and the training loss remained essentially flat despite extensive tuning of learning rates and regularization. This behavior is consistent with recent findings that embedding norms play a critical role in convergence and gradient scaling in representation learning [4]. Beyond optimization issues, the concatenation baseline remains within the single-vector retrieval paradigm and thus inherits its theoretical limitations [5].
>
> ---
>
> We sincerely appreciate your review in helping improve our paper and hope our response will be satisfactory. Please feel free to leave us more comments, and we will be happy to engage in discussions.
>
>
>
> References:
>
> [1] Douze, Matthijs, et al. "The faiss library." *IEEE Transactions on Big Data* (2025).
>
> [2] Scheerer, Jan Luca, et al. "WARP: An efficient engine for multi-vector retrieval." *Proceedings of the 48th International ACM SIGIR Conference on Research and Development in Information Retrieval*. 2025.
>
> [3] Wang, Yichuan, et al. "LEANN: A Low-Storage Vector Index." *arXiv preprint arXiv:2506.08276* (2025).
>
> [4] Draganov, Andrew, et al. "On the Importance of Embedding Norms in Self-Supervised Learning." *arXiv preprint arXiv:2502.09252* (2025).
>
> [5] Weller, Orion, et al. "On the theoretical limitations of embedding-based retrieval." *arXiv preprint arXiv:2508.21038* (2025).

---

> ### Comment · Reviewer_L8o6 · 2025-11-18
> **Thank you for your response!**
>
> > W1 Unclear generalization ability & Q2 Task-specific prompt setup
>
> 1. The reviewer persists that the current ID vs OOD evaluation is somewhat flawed. For example: ImageNet is in the training dataset but ObjectNet [1] is considered OOD, despite them having (113) overlapped classes. Visual7W [2] (OOD) is derived from MSCOCO (ID).
>
> 2. Reusing instructions between training and evaluation could result in overfitting/task leakage. E.g. ImageNet performance is predicated on a specifc instruction/phrase.
>
> However, the reviewer acknowledges these weaknesses are mostly inherited from standard (although in the reviewer's opinion) bad practices in the community. 1 is also somewhat mitigated by a careful reading of the task breakdown in the amended manuscript.
>
> > W2, W3, W4, W5, W6, Q2
>
> Thank you for adding additional details and discussion around limitations, especially the clarifications on the latency analysis.
>
> > All models reported in Tables 1 and 2 are trained on the same mixture of MMEB-train and ViDoRe-train data as indicated in Lines 303-305
>
> I am somewhat confused by this. To the reviewer's knowledge,  MetaEmbed, MoCa, GME are trained on ViDoRe-train, whereas VLM2Vec, SigLIP and mmE5 are not.
>
> Overall, I remain positive on the paper and will keep my score.
>
> ----
> [1] Barbu et al., ObjectNet: a large-scale bias-controlled dataset for pushing the limits of object recognition models. Proceedings of the 33rd International Conference on Neural Information Processing Systems. 2019.
>
> [2] Yuke Zhu, O. Groth, M. Bernstein, and L. Fei-Fei, “Visual7W: Grounded Question Answering in Images,” in 2016 IEEE Conference on Computer Vision and Pattern Recognition (CVPR), 2016.

---

> > ### Author Response · Authors · 2025-11-19
> > **Follow-up Response to Reviewer L8o6**
> >
> > Thank you for engaging in this meaningful discussion!
> >
> > ---
> >
> > > **W1 Unclear generalization ability & Q2 Task-specific prompt setup**
> >
> > We believe you exactly pointed out that the overlapped classes not considered OOD is a concern for the multimodal retrieval community. Thus, in light of your comments, we added further discussion in Appendix A.3 regarding the flaws in the current multimodal retrieval evaluation, and hopefully, we can advocate for a more systematic evaluation setup and ask for attention from the community. For instance, in image retrieval, it is becoming increasingly standard [1] to exclude overlapping classes when assessing generalization, and we believe the same principle should be adopted for multimodal retrieval going forward.
> >
> > ---
> >
> > > W2, W3, W4, W5, W6, Q2
> >
> > We are glad we addressed those concerns!
> >
> > ---
> >
> > > **All models reported in Tables 1 and 2 are trained on the same mixture of MMEB-train and ViDoRe-train data as indicated in Lines 303-305**
> >
> > We apologize for the confusion caused by our wording. To clarify: all **MetaEmbed** models are trained on the mixture of MMEB-train and ViDoRe-train.
> >
> > ---
> >
> > Thanks again for your quick response and keeping your positive recognition.
> >
> > Reference:
> >
> > [1] Song, Chull Hwan, et al. "On train-test class overlap and detection for image retrieval." *Proceedings of the IEEE/CVF Conference on Computer Vision and Pattern Recognition*. 2024.

---

### Author Response · Authors · 2025-11-18
**Rebuttal Revision and General Response to Reviewers**

Dear Reviewers and ACs,

Thank you for taking the time to review our paper! We are glad that all reviewers recognized the strengths of MetaEmbed and **gave positive initial feedback.**

During the rebuttal, we focus on addressing concerns that appeared across reviews and update the submission accordingly **with the differences highlighted in BLUE**:

- Generalization and clarity on training data / in-distribution vs out-of-distribution (reviewers L8o6, rgEq).
  - We emphasize the explicit *OOD* subset in MMEB and reported MetaEmbed’s OOD performance under this stricter protocol. We also clarified that all models in Tables 1 and 2 are trained on the same mixture of MMEB-train + ViDoRe-train.
- Sensitivity to MMR group sizes and justification of the chosen configuration (reviewers dVyq, rgEq).
  - We explain the design principles behind the group schedule {(1,1), (2,4), (4,8), (8,16), (16,64)}: exponential growth on the query side and higher capacity on the candidate side. We also added preliminary sensitivity evidence: extending the schedule with an additional large group (32,128) does not improve validation performance but increases training and memory cost, indicating diminishing returns beyond our chosen configuration.
- Efficiency and scalability considerations for practical deployment.
  - We clarify the concrete inference setup for Table 3 (PyTorch tensor index + `torch.einsum` scoring), stated that these measurements are a conservative upper bound rather than an optimized production system, and discussed how practical deployments would rely on FAISS or multi-vector-specific indexes plus standard techniques to handle large candidate pools and GPU memory constraints.

We hope these revisions and clarifications address the shared concerns while strengthening the overall narrative of MetaEmbed as a practical, general, and extensible solution for scalable multimodal retrieval.

---

### Meta-Review · Area_Chair_SaVL · 2026-01-02

**Summary:**

This paper received unanimously positive reviews. The reviewers recognized the novel and well-motivated idea, its versatile implementation, extensive experiments, strong results of the paper. They also raised diverse concerns with suboptimal evaluation of the generalization capability of the model (L8o6), incremental novelty over Matryoshka representation learning (rgEq), lack of empirical analysis (e.g., latency (L8o6), interpretability and qualitative results (dVyq), impact of embedding granularity (g6en), efficiency in training (rgEq), impact of the group size (rgEq), etc.), unclear description of the model used in experiments (L8o6), lack of details for reproducing the results (L8o6), potential complexity issues with a large candidate pool (L8o6), and missing discussions and comparisons with relevant work (L8o6, g6en). The AC found that most of these concerns have been well addressed by the thorough rebuttal and revision, and thus recommend acceptance of the paper. The authors are highly encouraged to add all the new results and discussions in the camera-ready version of the paper.

**Reviewer Concerns:**

All the concerns raised by the reviewers have been well addressed.
- Suboptimal evaluation of the generalization capability of the model (L8o6)
	- *The rebuttal assuaged this concern to some extent; the reviewer still believes the evaluation protocol is flawed, but acknowledges that it is inherited from the standard practices in the community. The authors further clarified the limitations of the protocol in the revision to reflect the reviewer's opinion.* **_The reviewer was satisfied by the authors' response and finally kept their positive rating._**
- Unclear description of the model used in experiments (L8o6)
	- *The models used in the experiments are elaborated on in the rebuttal.*
- Lack of details for reproducing the results (L8o6)
	- *Sufficient details were added to the revision.*
- Limited analysis on the latency (L8o6)
	- *Details of the existing analysis are provided in the revision.*
- Leading to high complexity when the candidate pool is large (L8o6)
	- *This issue is common in most vector-based retrieval systems, and the limitation has been discussed clearly in the revision.*
- Missing discussions and comparisons with relevant work (L8o6, g6en)
	- *These concerns have been well addressed by the rebuttal and reflected in the revision.*
- Lack of interpretability and qualitative analysis (dVyq)
	- *Qualitative results were added and discussed in the revision.*
- Lack of analysis on the impact of embedding granularity (g6en)
	- *Fully assuaged by the rebuttal with additional empirical analysis.*
- Incremental novelty over Matryoshka Representation Learning (MRL) (rgEq)
	- *The authors' response to this comment sounds reasonable: although they share similar ideas in concept level, the proposed method is substantially different from MRL in multiple aspects due to the different targets.*
- Insufficient analysis on efficiency in training (rgEq)
	- *Additional results in the rebuttal fully resolve this concern.*
- Lack of analysis on the impact of the group size (rgEq)
	- *Addressed by the rebuttal to some extent, but more analysis may be beneficial to better support the authors' argument.*

**Reviewer Scores:**

L8o6 is satisfied by the response and remains positive. No other reviewer engaged in the discussion, but their initial ratings were all positive and most of their concerns have been well addressed by the rebuttal. The AC thus expects that the reviewers would keep or increase their positive ratings at the end of the discussion period.

---

### Decision · Program_Chairs · 2026-01-26

Accept (Oral)